# Effect of Cold Plasma Treatment on the Packaging Properties of Biopolymer-Based Films: A Review

Monjurul Hoque [1,2], Ciara McDonagh [1], Brijesh K. Tiwari [3], Joseph P. Kerry [2] and Shivani Pathania [1,*]

1   Food Industry Development Department, Teagasc Food Research Centre, Ashtown,
    D15 KN3K Dublin, Ireland; Monjurul.Hoque@teagasc.ie (M.H.); Ciara.McDonagh@teagasc.ie (C.M.)
2   School of Food and Nutritional Sciences, University College Cork, T12 R229 Cork, Ireland; Joe.kerry@ucc.ie
3   Food Chemistry and Technology Department, Teagasc Food Research Centre, Ashtown,
    D15 KN3K Dublin, Ireland; Brijesh.Tiwari@teagasc.ie
*   Correspondence: Shivani.Pathania@teagasc.ie

**Abstract:** Biopolymers, like polysaccharides and proteins, are sustainable and green materials with excellent film-forming potential. Bio-based films have gained a lot of attention and are believed to be an alternative to plastics in next-generation food packaging. Compared to conventional plastics, biopolymers inherently have certain limitations like hydrophilicity, poor thermo-mechanical, and barrier properties. Therefore, the modification of biopolymers or their films provide an opportunity to develop packaging materials with desired characteristics. Among different modification approaches, the application of cold plasma has been a very efficient technology to enhance the functionality and interfacial characteristics of biopolymers. Cold plasma is biocompatible, shows uniformity in treatment, and is suitable for heat-sensitive components. This review provides information on different plasma generating equipment used for the modification of films and critically analyses the impact of cold plasma on packaging properties of films prepared from protein, polysaccharides, and their combinations. Most studies to date have shown that plasma treatment effectively enhances surface characteristics, mechanical, and thermal properties, while its impact on the improvement of barrier properties is limited. Plasma treatment increases surface roughness that enables surface adhesion, ink printability, and reduces the contact angle. Plasma-treated films loaded with antimicrobial compounds demonstrate strong antimicrobial efficacy, mainly due to the increase in their diffusion rate and the non-thermal nature of cold plasma that protects the functionality of bioactive compounds. This review also elaborates on the existing challenges and future needs. Overall, it can be concluded that the application of cold plasma is an effective strategy to modify the inherent limitations of biopolymer-based packaging materials for food packaging applications.

**Keywords:** cold plasma; biopolymers; packaging; antimicrobial; biodegradable; modification

## 1. Introduction

The packaging of food materials provides protection, assists shelf-life, and facilitates both communication to consumers and the transportation of food products. Materials such as paper, plastic, metal, and glass are predominantly used in food packaging. Plastic is the primary packaging material used in the food industry due to its light weight, flexibility, high performance, and low cost [1]. The proliferation of plastic products in the last few decades has been extraordinary and in 2019 global plastic production was at 368 million metric tons and continuing to grow by 4% every year [1,2]. Packaging is the largest end-use market segment of single-use plastics and it accounts for over 40% of total plastic usage [3].

Despite the practical benefits, the downsides of using plastic and the mismanagement of plastic waste have become critical for individuals, communities, and the whole ecosystem [4,5]. Of all the plastic produced, 50% is for single-use purposes; less than 9% of all plastics gets recycled; 10 million tons of plastics are dumped in the ocean every

year; around 1 million marine animals are killed every year [3], and currently the ocean is expected to contain more plastic than fish by the year 2050 [6]. In a study by the University of Newcastle, Australia, researchers revealed the impact of plastic pollution on humans and reported that an average person may ingest about 5 g of plastic every week [7].

Therefore, immediate tactical action and strategic adjustment across the plastic trade chain and its life cycle are needed to stop leakage and the further accumulation of plastic in nature. Among the other strategies, banning problematic single-use plastics has the potential to lower plastic use up to 40% by the year 2030 [8]. The packaging sector is the dominant generator of plastic waste and in 2015 it produced almost half the global plastic waste [9], as shown in Figure 1; of which the food packaging industry accounts for 50% of plastic use [10]. Therefore, packaging has been under scrutiny as it constantly produces a high amount of plastic waste and this has forced researchers and the industry to consider sustainable alternatives [10,11].

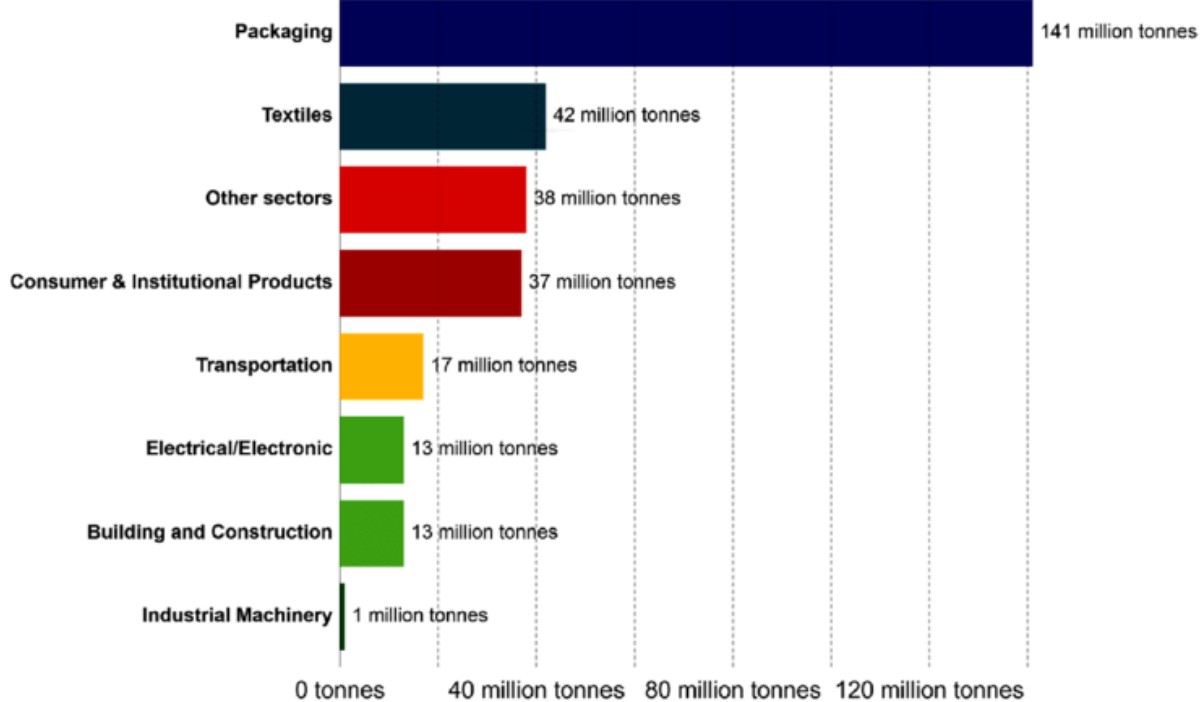

**Figure 1.** Global plastic waste generation by industrial sector, measured in tons per year, Reprinted from Ref. [9].

Bioplastics have received significant attention owing to their application as one of the most promising sustainable alternatives to replace synthetic plastics [10,12–15]. The demand for bioplastics has augmented over the years as these polymers are nontoxic, biocompatible, renewable, and biodegradable [4,16]. In 2020, global bioplastic production capacity was 2.11 million tons, which is only about 1% of the total plastic and set to increase to approximately 2.87 million tons in the next five years [17]. Of the total bioplastic produced, 58.1% is biodegradable and there are more than 20 groups of these biodegradable bioplastics [18]. However, only three groups of biopolymers are being commercially explored and they account for 95% of the production capacity as shown in Figure 2: (a) starch blend; (b) polybutylene (polybutylene succinate (PBS), and polybutylene adipate terephthalate (PBAT)); and (c) PLA [18]. Other biopolymers such as polysaccharide-based cellulose [19], sodium alginate [20], pectin [21], zein [22], carrageenan [23], chitosan [24], protein-based: casein [25], whey protein [26], gelatin [27], and keratin [28] are also being explored to develop packaging materials. However the biopolymer-based packaging materials have certain shortcomings due to their inherent properties like hydrophilicity, the high risk of microbial contamination, poor barrier, and mechanical properties, low thermal stabil-

ity, low surface functionality, poor printability, and adhesiveness when compared with conventional plastics [29–35].

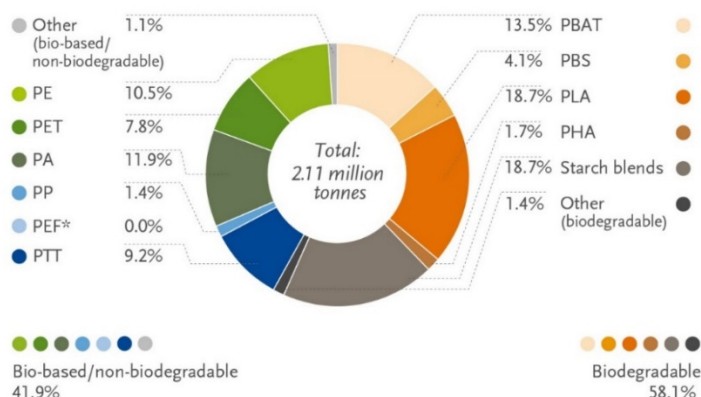

**Figure 2.** Global bioplastic production capacities 2020 by material type, Reprinted from Ref. [17].

To overcome the aforementioned limitations of neat biopolymer-based films, several modification techniques are being applied, such as: the use of cross-linkers like glutaraldehyde, citric acid, calcium chloride, formaldehyde [33,36,37]; the addition of additives like plasticizers; functional components like essential oils; and antimicrobial compounds such as nisin and lysozymes [35,38,39]. Other techniques include hydrophobic modification with structured oil nanoparticles [29,40–42], incorporating organic and inorganic fillers [30], coating biopolymers with synthetic polymers, or a combination of two or more biopolymers having potential packaging properties [43–46]. The application of novel technologies like high-pressure processing, cold plasma, ultrasonication, high-pressure homogenization, irradiation, and microwaves [47] have also been examined.

Cold plasma treatment is a novel technique that modifies the surface of the polymer providing opportunities to develop packaging materials with desired characteristics. For instance, the application of a cold plasma-activated polylactic acid surface facilitated the coating of nisin, resulting in the formation of antimicrobial packaging materials [48]. The grafting of polycaprolactone on the plasma-treated zein film showed enhanced water barrier properties [49]. Coating of plasma-treated low-density polyethylene with carboxymethyl cellulose or collagen loaded with cinnamaldehyde showed improved physical, mechanical, antioxidant, and antimicrobial activities suitable for packaging tilapia fillets [45]. The pre-treatment of zein and polyacid with cold plasma amplified the compatibility and adhesion between the layers and enhanced the mechanical, water barrier, and hydrophobic properties of the films [46].

Recently a number of studies have been carried out examining the application of cold plasma in the modification of biopolymer-based packaging films. Therefore, the main objective of this paper is to undertake a systematic review of the application of plasma for the modification of packaging materials suitable for food packaging applications as outlined in scientific literature. Section 2 of this review discusses the production of plasma and features prominent plasma-producing devices. Section 3 critically analyses the impact of cold plasma on surface roughness, contact angle, molecular properties, mechanical properties, thermal properties, barrier properties, antimicrobial effects, and biodegradability. Section 4 discusses safety concerns in relation to the application of cold plasma in the preparation of films used for food packaging applications.

*Search Strategy and Selection of Articles*

A systematic search was carried out until 31 November 2021, with the help of the keywords "plastic waste", "biopolymer", "bioplastic", "film", "biodegradable", "novel technology", "plasma", "cold plasma", "dielectric barrier discharge", along with the Boolean operators AND, OR and NOT. Google Scholar, Science Direct, Scopus database, and Web of

Science were used to obtain the desired information. All the articles were studied carefully, and after a systematic review of the carefully chosen literature, the results of the authors were compared and considered until a consensus was reached.

## 2. Plasma

Plasma is referred to as the fourth state of matter and is usually visualized as an arc or discharge of bright fluorescent light. Plasma in its physical form is found to be a partially or completely ionized gas mixture of reactive species like electrons, photons, positively or negatively charged ions, free radicals, and gas molecules or atoms in their excited or fundamental states with a net neutral charge. Thus plasma is considered as "quasi-neutral" [50–52]. Plasma is induced under varying processing temperatures and pressures by energizing a neutral gas, thus being classified into thermal (equilibrium) plasma and non-thermal (non-equilibrium) plasma [50,52,53]. The thermal plasma requires a high pressure (>105 Pa) and power supply (power > 50 MV), in which the main feature is the existence of thermodynamic equilibrium among all generated species at a temperature range of $2 \times 10^3$ K to $3 \times 10^4$ K. In contrast, the non-thermal plasma does not require high pressure or power and can exist without a localized thermodynamic equilibrium, thus referred to as non-equilibrium plasma. It is further subdivided into two categories namely (i) quasi-equilibrium plasma (100–150 °C), where the generated species are in a local thermodynamic equilibrium state, and (ii) non-equilibrium plasma (<60 °C) where electrons possess a higher temperature and heavier species have a moderate temperature without any local thermodynamic equilibrium, thereby rendering a lower temperature to the whole system. This low temperature, non-equilibrium plasma is also known as cold plasma [50,52–55].

Cold plasma, also referred to as atmospheric pressure plasma, is generated upon the application of an electrical field to a neural gas at atmospheric pressure. Either direct current (DC) or alternate current (AC) at a frequency (ranging between Hz to GHz) is applied to the gas or gas mixture between two electrode plates. The plasma generation from the neutral gas is associated with the three major phenomena that are excitation, ionization, and dissociation. Neutral gas or a mixture of gases contains few carriers in the form of electrons or ions. Upon application of an electric field, free carrier charges get accelerated and may collide with ions, or atoms in the gas or electrode surface. The excitation process increases the translational and transitional energy of the atoms. Therefore, sufficient energy in the excitation process results in the ionization of atoms by removing their loosely bound electrons. Both excitation and ionization phenomena may be due to the collision within and between different gaseous components (electrons, ions, and atoms, neutral molecules) and radiations. On the other hand, the dissociation is a result of an inelastic collision of a molecule with an electron, ion, or photon [53,55,56]. Over the years, plasma technology has been used in the area of food and biomedical applications. Cold plasma is an emerging, green technology that presents various potential applications in sustainable biopolymer-based food packaging. Notably, the application of cold plasma for surface functionalization of polymers by the introduction of selective functional groups allows the modification of surface properties giving improved wettability, sealability, and printability, promoting adhesion without compromising the desired bulk properties of the polymer [52,57]. It has been found that gas plasma reactions improved the mechanical properties of biopolymer-based films by enhancing the polymer resistance towards failure [46,58]. It also improved the barrier property of the film against gases by plasma deposition of a barrier layer [59,60].

Depending on the excitation frequency, atmospheric pressure plasma or cold plasma may be classified as DC and low-frequency discharges (corona discharges and dielectric barrier discharges); radiofrequency discharges (atmospheric pressure plasma jets-APPJ); and microwave driven plasma. Table 1 shows some of the prominent cold plasma generating systems and their features.

**Table 1.** The prominent cold plasma generating equipment.

| Atmospheric Cold Plasma Source | Device Diagram | Features | Advantage/Disadvantage/Application | References |
|---|---|---|---|---|
| Corona Discharge |  | ❖ Corona discharge has a pair of inhomogeneous electrodes, discharge produced is weakly luminous and always spatially non-uniform and dependent on the geometry of the setup. <br> ❖ It has a localized emission and discharges primarily near sharp points, edges, or around thin wires with a relatively high electric field at atmospheric pressure. <br> ❖ The volume affected by corona discharge is expansively lower when compared to total corona discharge volume. | Disadvantages: <br> ❖ Non-uniform discharge and causes power loss. <br> ❖ The treatment surface is very small due to the production of the small volume of discharge. <br><br> Advantages: <br> ❖ Ease of operation. <br><br> Application: <br> ❖ water purification; production of ozone; treatment of the material surface, gaseous contamination; removal of undesirable volatile organic substances present in the atmosphere; inactivation of microorganisms; modification of physicochemical properties of proteins and starch, surface treatment of packaging materials. | [50,56,61–67]. Reprinted from Ref. [65]. |
| Dielectric barrier discharges (DBD) |  | ❖ In general, DBD plasma configuration can be planar or cylindrical having two electrodes, of which one of the electrodes is covered with the dielectric barrier (insulating material) and the other is grounded. <br> ❖ The insulating materials can be glass, quartz, ceramics, plastics, and silicon rubber. <br> ❖ DBD Plasma is generated in the gas or the mixture of the gases present in between the electrodes gap. <br> ❖ DBD is obtained at high voltage and frequencies in the range of (1 to 100 kV) and 0.05 to 500 kHz. <br> ❖ Based on the geometry and the set-up of electrodes, DBD exhibits two different modes: Filamentary discharge (non-uniform) and diffuse mode (homogeneous discharge). | Disadvantages: <br> ❖ Required high voltage for ignition. <br> ❖ Narrow discharge gap height and has a direct impact on plasma homogeneity. <br><br> Advantages: <br> ❖ Low energy consumption as compared to corona discharge <br> ❖ High effectiveness, easily scalable, low operational cost, and short processing time. <br> ❖ Can be designed as per processing requirements like sliding discharge, capillary plasma electrode discharge, microcavity plasma array, and coplanar configurations. <br><br> Application: <br> ❖ Production of ozone; waste-water purification; inactivate microbes; surface modification; excitation of $CO_2$ lasers and excimer lamp; plasma chemical vapour deposition, treatment of cancer, dentistry, as well as in the field of regenerative medicines. | [56,63,68–70] Reprinted from Ref. [56]. |

**Table 1.** *Cont.*

| Atmospheric Cold Plasma Source | Device Diagram | Features | Advantage/Disadvantage/Application | References |
|---|---|---|---|---|
| Atmospheric pressure plasma jets (APPJ) | 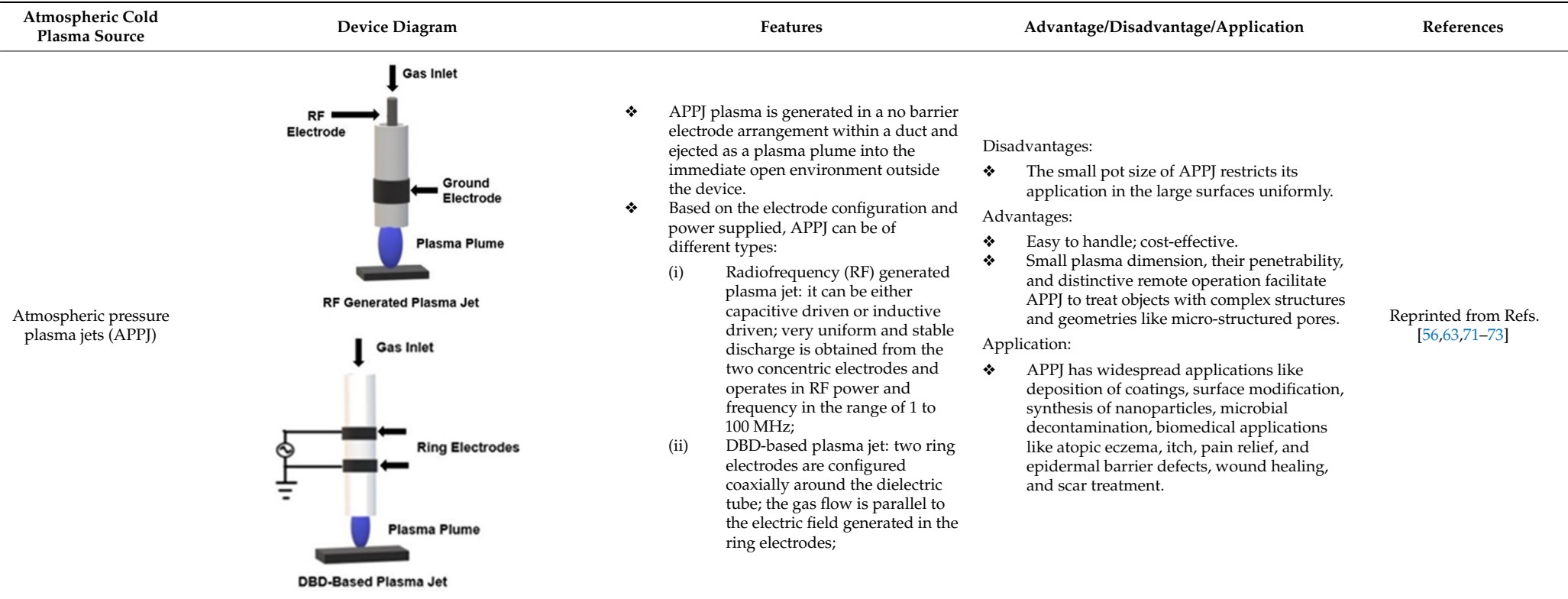 | ❖ APPJ plasma is generated in a no barrier electrode arrangement within a duct and ejected as a plasma plume into the immediate open environment outside the device.<br>❖ Based on the electrode configuration and power supplied, APPJ can be of different types:<br>(i) Radiofrequency (RF) generated plasma jet: it can be either capacitive driven or inductive driven; very uniform and stable discharge is obtained from the two concentric electrodes and operates in RF power and frequency in the range of 1 to 100 MHz;<br>(ii) DBD-based plasma jet: two ring electrodes are configured coaxially around the dielectric tube; the gas flow is parallel to the electric field generated in the ring electrodes; | Disadvantages:<br>❖ The small pot size of APPJ restricts its application in the large surfaces uniformly.<br>Advantages:<br>❖ Easy to handle; cost-effective.<br>❖ Small plasma dimension, their penetrability, and distinctive remote operation facilitate APPJ to treat objects with complex structures and geometries like micro-structured pores.<br>Application:<br>❖ APPJ has widespread applications like deposition of coatings, surface modification, synthesis of nanoparticles, microbial decontamination, biomedical applications like atopic eczema, itch, pain relief, and epidermal barrier defects, wound healing, and scar treatment. | Reprinted from Refs. [56,63,71–73] |

**Table 1.** *Cont.*

| Atmospheric Cold Plasma Source | Device Diagram | Features | Advantage/Disadvantage/Application | References |
|---|---|---|---|---|
| | 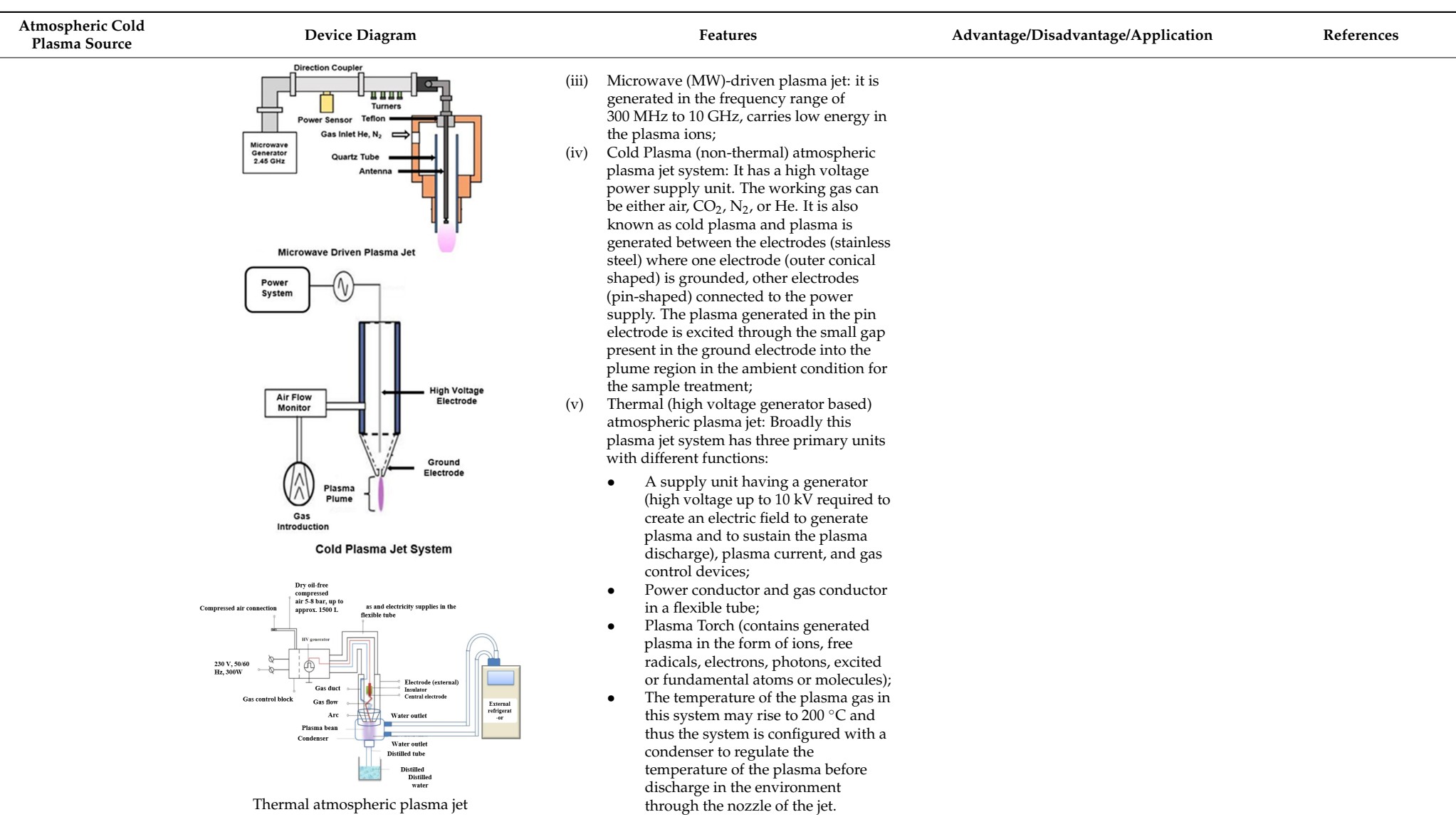<br><br>Microwave Driven Plasma Jet<br><br>Cold Plasma Jet System<br><br>Thermal atmospheric plasma jet | (iii) Microwave (MW)-driven plasma jet: it is generated in the frequency range of 300 MHz to 10 GHz, carries low energy in the plasma ions;<br>(iv) Cold Plasma (non-thermal) atmospheric plasma jet system: It has a high voltage power supply unit. The working gas can be either air, $CO_2$, $N_2$, or He. It is also known as cold plasma and plasma is generated between the electrodes (stainless steel) where one electrode (outer conical shaped) is grounded, other electrodes (pin-shaped) connected to the power supply. The plasma generated in the pin electrode is excited through the small gap present in the ground electrode into the plume region in the ambient condition for the sample treatment;<br>(v) Thermal (high voltage generator based) atmospheric plasma jet: Broadly this plasma jet system has three primary units with different functions:<br>• A supply unit having a generator (high voltage up to 10 kV required to create an electric field to generate plasma and to sustain the plasma discharge), plasma current, and gas control devices;<br>• Power conductor and gas conductor in a flexible tube;<br>• Plasma Torch (contains generated plasma in the form of ions, free radicals, electrons, photons, excited or fundamental atoms or molecules);<br>• The temperature of the plasma gas in this system may rise to 200 °C and thus the system is configured with a condenser to regulate the temperature of the plasma before discharge in the environment through the nozzle of the jet. | | |

## 3. Impact of Cold Plasma on Packaging Properties

### 3.1. Surface Roughness

The application of cold plasma increases the surface roughness of biopolymer films and can be attributed to the etching effect phenomena. This is caused by the bombardment of high-energy plasma species like electrons, photons, positively or negatively charged ions, free radicals, and gas molecules or atoms in their excited or fundamental states [74,75]. The etching effect in a film's surface may arise due to chemical processes such as cleavage of chemical bonds, scission of polymer chains, or chemical degradation of film components by the influence of the free radicals. It can also be caused by a physical process like the removal or re-aggregation of low molecular weight components/fragments on the surface of the film. The other possible reason for the change in the surface roughness by the application of cold plasma may be due to its micro-discharge filaments [43,52,75].

The extent of roughness may vary depending on the power supplied (voltage), exposure time, and uniformity of the exposed energy of the plasma species onto the film surface [52,58]. The etching effect phenomenon is noticed commonly with the application of DBD plasma, radiofrequency (RF) generated plasma, microwave (MW)-driven plasma with different gases like air, argon, $O_2$, $N_2$, and $CO_2$. The overall variation in the surface roughness of the film exposed to cold plasma is summarized in Table 2.

As shown in Figure 3I, the Atomic Force Microscopic (AFM) topography of the zein film treated with DBD atmospheric cold plasma demonstrated that the roughness of the polymer surface increased with the increase in voltage and treatment time. A 5 min treatment at a voltage of 60, 70, and 80 kV increased the roughness (RRMS: Root Mean Square Roughness) from 16.41 nm to 52.04, 126.16, and 132.87 nm, respectively. This enhancement in the roughness was attributed to the etching effect of plasma treatment [75]. Such incremental increases in surface roughness were also observed for polylactic acid/nisin film treated with cold plasma, where the surface roughness value of the untreated film was 4.26 nm and increased to 6.24 nm with the increase in treatment time from 0 to 30 s. A further increase in treatment time led to a decrease in surface roughness and this could be due to the longer treatment time (the 60 s) having etched off the PLA microscale protuberances and the appearance of nano-needle-like bumps, resulting in a more uniform PLA surface [48]. Also, Cools, Asadian [76] treated biodegradable polyethylene-oxide terephthalate (PEOT)/polybutylene terephthalate (PBT) film with DBD plasma involving air, He, Ar, and $N_2$. AFM topography showed $N_2$ and air plasma treatment of PEOT/PBT film increased the surface roughness whereas He, and Ar plasma treatment decreased the surface roughness as compared to untreated sample as shown in Figure 3II. It was reported that He and Ar plasma treatment etched away the sharp peaks present on the surface of the PEOT/PBR film resulting in a more uniform surface. No significant change in the surface topography or surface roughness was observed for $N_2$ treated PEOT/PBR film and the application of air plasma completely changed the surface topography by forming hemispherical protrusions on the film surface. Such variation in etching behavior can be attributed to the difference in active species present in the plasma discharge. Notably, air plasma containing significantly higher oxygen molecules created an atomic oxygen species that is known for excellent etching phenomena [76]. Therefore, it is clear that the variation in surface roughness is a function of four factors: voltage; treatment time; types of plasma gas; and the internal structure of the polymer.

**Table 2.** Effect of cold plasma on packaging properties of biopolymer-based films. Reprinted from Refs. [77,78]. Reprinted with permission from Refs. [46,58,79–85], Elsevier, 2022. Reprinted with permission from Refs. [49,75,86,87], Wiley Online Library, 2022.

| Types of Biopolymers | Device Diagram | Processing Conditions | Film Properties | | | | | | | | | | | R |
|---|---|---|---|---|---|---|---|---|---|---|---|---|---|---|
| | | | TS | EAB | CA | SR | WVP | OP | Tg | Tm | ΔH | Tdeg | B.D | |
| Corn starch (28% amylose) |  DBD | 20 kV, 200 Hz, 10, 15, and 20 min | ↑ | ↓ | ↑ | ↑ | = | – | ↓ | ↓ | ↓ | ↑ | – | [58] |
| Corn Starch/poly(ε-caprolactone) |  RF Generated Plasma Reactor | 40 W, 13.56 MHz, 2 min | – | – | ↓ | ↑ | ↑ | ↑ | – | – | – | – | ↑ | [79] |
| Chitosan and zein |  DBD Plasma Reactor | 65 V, 1.5 A, 30 s | – | – | ↓ | ↑ | ↓ | – | ↑ | – | – | – | – | [81] |

**Table 2.** *Cont.*

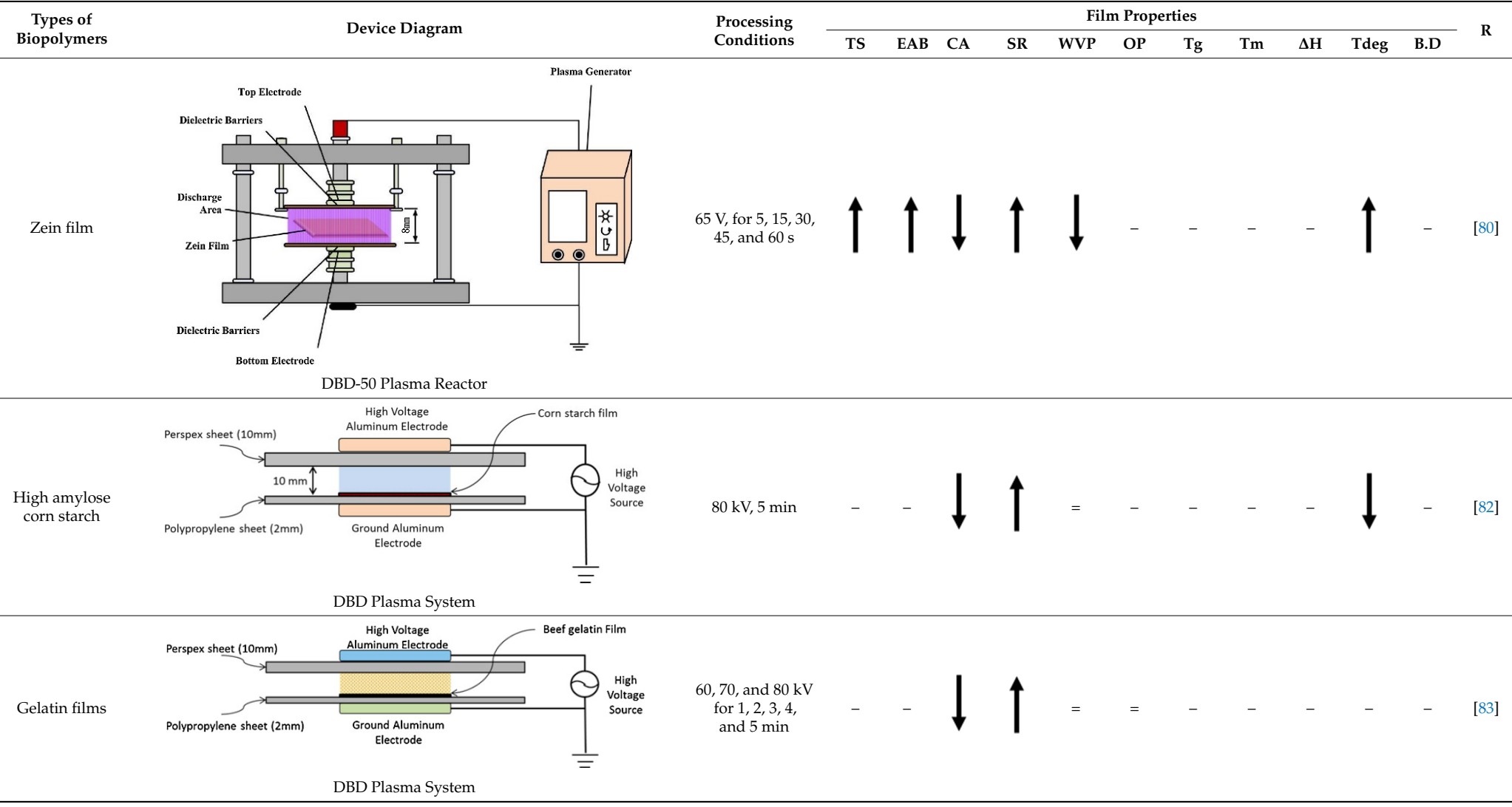

| Types of Biopolymers | Device Diagram | Processing Conditions | TS | EAB | CA | SR | WVP | OP | Tg | Tm | ΔH | Tdeg | B.D | R |
|---|---|---|---|---|---|---|---|---|---|---|---|---|---|---|
| Zein film | DBD-50 Plasma Reactor | 65 V, for 5, 15, 30, 45, and 60 s | ↑ | ↑ | ↓ | ↑ | ↓ | – | – | – | – | ↑ | – | [80] |
| High amylose corn starch | DBD Plasma System | 80 kV, 5 min | – | – | ↓ | ↑ | = | – | – | – | – | ↓ | – | [82] |
| Gelatin films | DBD Plasma System | 60, 70, and 80 kV for 1, 2, 3, 4, and 5 min | – | – | ↓ | ↑ | = | = | – | – | – | – | – | [83] |

**Table 2.** *Cont.*

| Types of Biopolymers | Device Diagram | Processing Conditions | Film Properties | | | | | | | | | | | R |
|---|---|---|---|---|---|---|---|---|---|---|---|---|---|---|
| | | | TS | EAB | CA | SR | WVP | OP | Tg | Tm | ΔH | Tdeg | B.D | |
| Chitosan |  DBD Plasma Treatment | 70 kV, 5 min | – | – | – | ↑ | – | – | – | – | – | = | – | [77] |
| Chitosan |  Plasma | 60, 70 and 80 kV for 1, 2, 3, 4 and 5 min | – | – | ↓ | ↑ | = | = | – | – | – | – | – | [86] |
| Starch (rice, potato, tapioca, corn) |  DBD Plasma | 80 kV for 5 min | – | – | – | ↑ | – | – | ↑ | – | – | – | – | [87] |
| Sodium caseinate |  DBD Plasma | 60 and 70 kV for 1, 2, 3, 4, and 5 min | – | – | – | ↑ | = | = | ↓ | – | – | – | – | [84] |

**Table 2.** *Cont.*

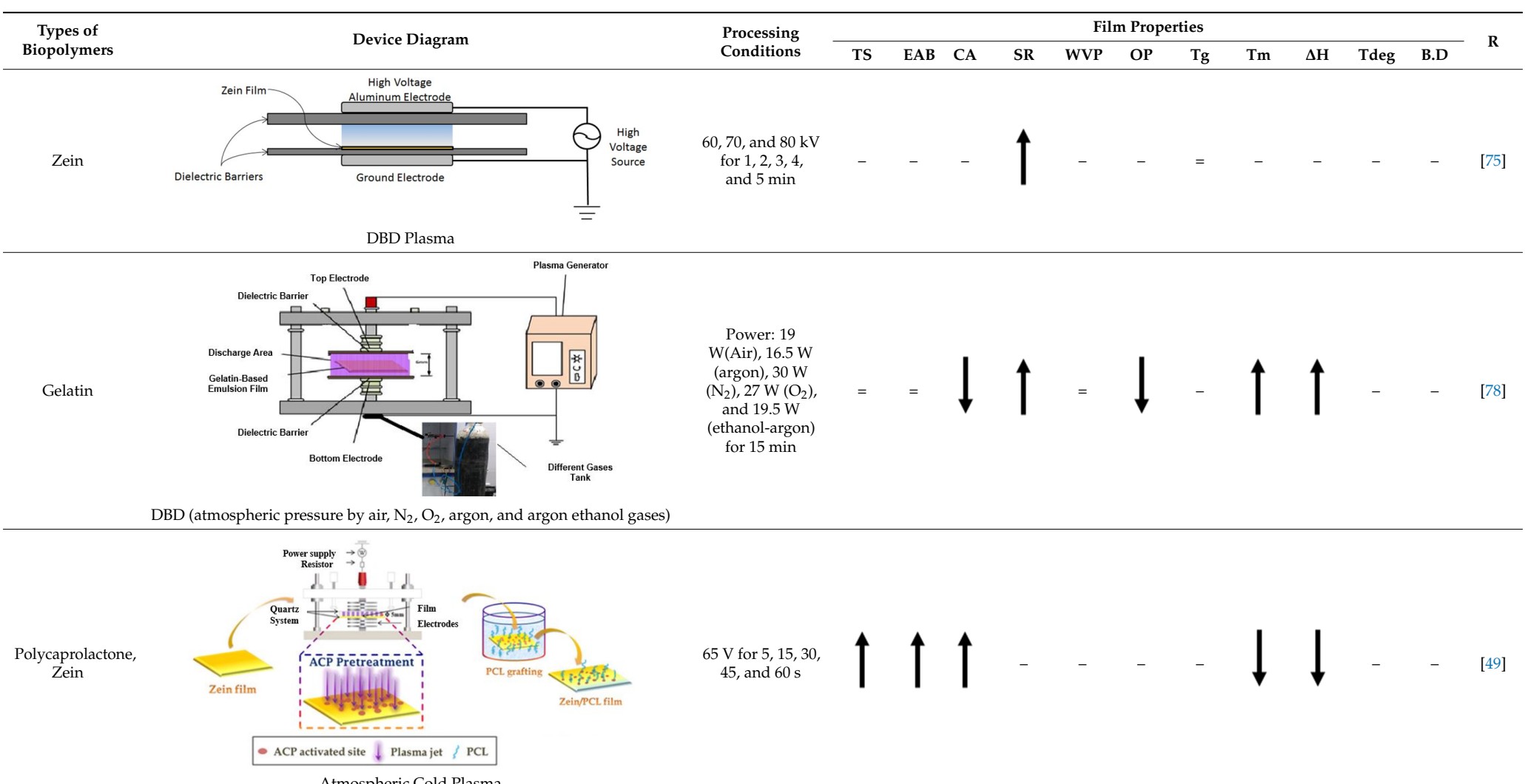

| Types of Biopolymers | Device Diagram | Processing Conditions | Film Properties | | | | | | | | | | | R |
|---|---|---|---|---|---|---|---|---|---|---|---|---|---|---|
| | | | TS | EAB | CA | SR | WVP | OP | Tg | Tm | ΔH | Tdeg | B.D | |
| Zein | DBD Plasma | 60, 70, and 80 kV for 1, 2, 3, 4, and 5 min | – | – | – | ↑ | – | – | = | – | – | – | – | [75] |
| Gelatin | DBD (atmospheric pressure by air, $N_2$, $O_2$, argon, and argon ethanol gases) | Power: 19 W(Air), 16.5 W (argon), 30 W ($N_2$), 27 W ($O_2$), and 19.5 W (ethanol-argon) for 15 min | = | = | ↓ | ↑ | = | ↓ | – | ↑ | ↑ | – | – | [78] |
| Polycaprolactone, Zein | Atmospheric Cold Plasma | 65 V for 5, 15, 30, 45, and 60 s | ↑ | ↑ | ↑ | – | – | – | – | ↓ | ↓ | – | – | [49] |

**Table 2.** *Cont*.

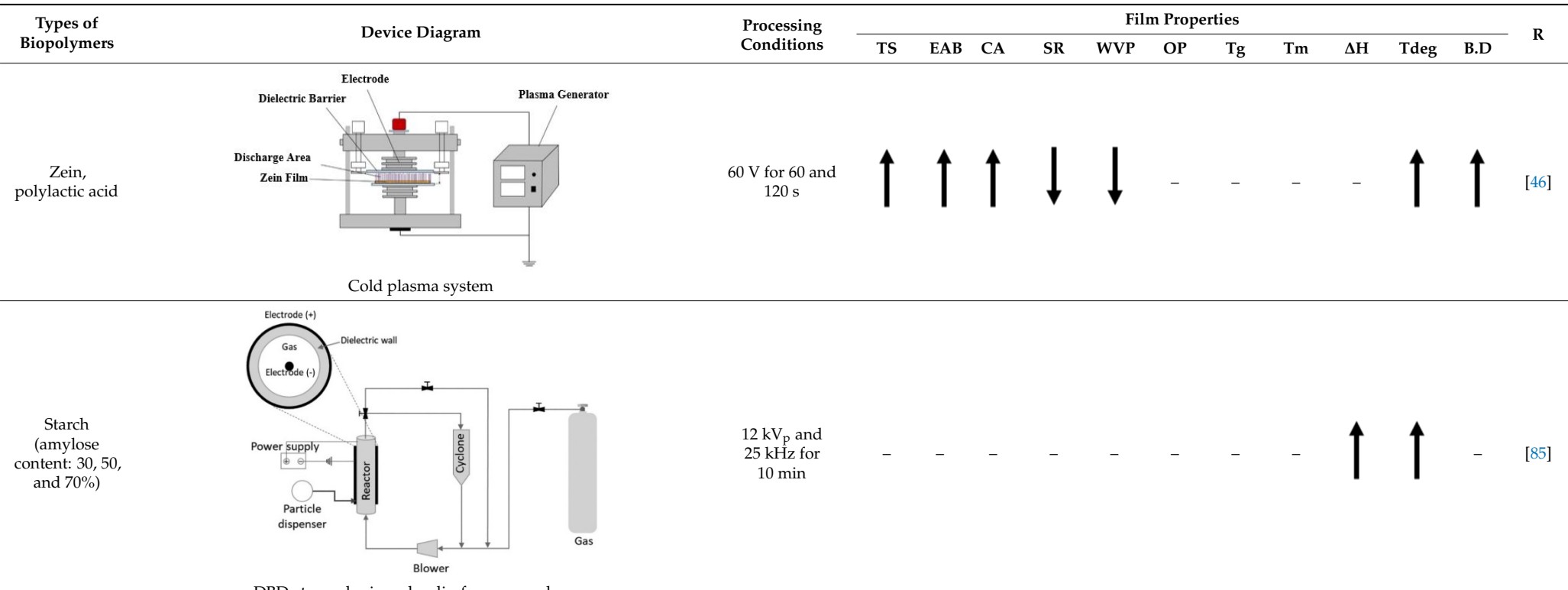

| Types of Biopolymers | Device Diagram | Processing Conditions | Film Properties | | | | | | | | | | | R |
|---|---|---|---|---|---|---|---|---|---|---|---|---|---|---|
| | | | TS | EAB | CA | SR | WVP | OP | Tg | Tm | ΔH | Tdeg | B.D | |
| Zein, polylactic acid | Cold plasma system | 60 V for 60 and 120 s | ↑ | ↑ | ↑ | ↓ | ↓ | – | – | – | – | ↑ | ↑ | [46] |
| Starch (amylose content: 30, 50, and 70%) | DBD atmospheric and radio-frequency plasma | 12 kV$_p$ and 25 kHz for 10 min | – | – | – | – | – | – | – | – | ↑ | ↑ | – | [85] |

TS: Tensile Strength; EAB: Elongation at Break; CA: Contact Angle; SR: Surface Roughness; WVP: Water Vapor Permeability; OP: Oxygen Permeability; T$_g$: Glass Transition Temperature; T$_m$: Melting Temperature; ΔH: Change in Enthalpy; T$_{deg}$: Thermal Degradation; BD: Biodegradation; R: Reference; ↑: Significant Increase in the Value With Compared to the Untreated Sample; ↓: Significant Decrease in the Value With Compared to the Untreated Sample; =: No Significant Change in the Value With Compared to the Untreated Sample; –: No Data Reported.

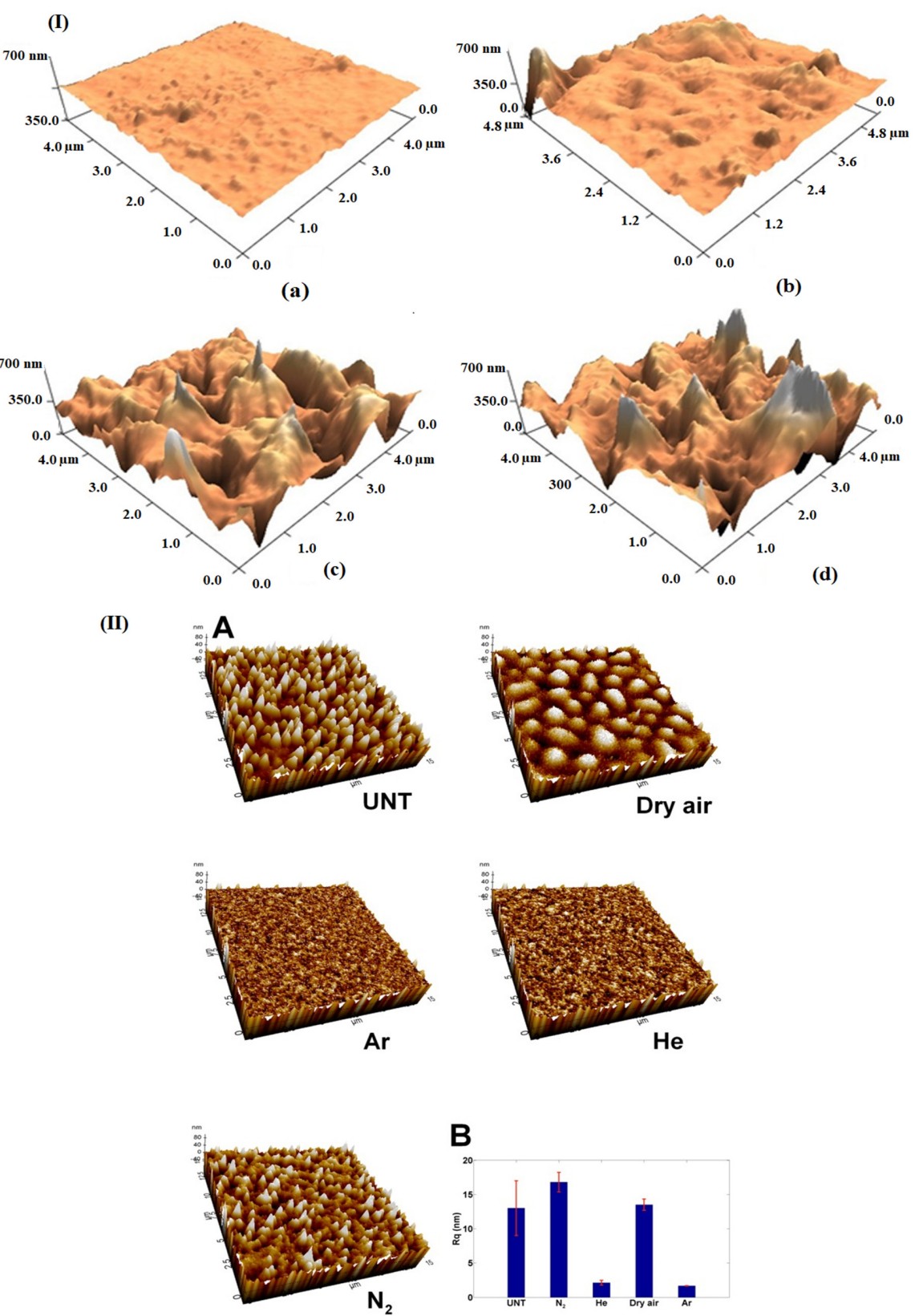

**Figure 3.** (**I**) AFM images of the surface of untreated and DBD plasma-treated zein films showing the impact of variation in voltage: (**a**) untreated, (**b**) 60 kV-5 min, (**c**) 70 kV-5 min, (**d**) 80 kV-5 min. Reprinted with permission from Ref. [75], Wiley Online Library, 2022; (**II**) AFM micrographs of untreated (UNT) and Dry air, Ar, He, and $N_2$ plasma-treated surface of the PEOT/PBT films; Bar chart shows the variation in root mean square roughness ($R_q$). Reprinted from Ref. [76].

Figure 4 shows the etching effect influencing another physical property of the film i.e., diffusion. The increased surface roughness of the films shows an enhanced diffusion coefficient of the reactive ingredients present in the film matrix when encountered with an aqueous solution. For instance, the application of DBD plasma into chitosan/thymol antimicrobial films showed increased diffusion of thymol when in contact with water [77]. This is due to the increase in roughness as a result of the etching effect that decreased the thickness and thus increased the diffusion rate. Similarly, an increase in thymol diffusion co-efficient was noticed in zein/thymol film upon application of DBD plasma [88]. In another study, Karam, Casetta [89] showed that cold plasma treatment modified the surface of low-density polyethylene films allowing the controlled absorption of nisin leading to the development of active packaging film [89]. Similar to the diffusion phenomena, plasma treatment also influences the adsorption characteristic of polymer surfaces and it varies based on type plasma treatment. Karam, Jama [90] reported that exposure of LDPE to different plasma ($N_2$, and $Ar/O_2$) influenced the nisin adsorption. The untreated and $N_2$ plasma treated films showed lower adsorption of nisin as compared to $Ar/O_2$ plasma treated films, which could be attributed to higher surface roughness of LDPE film caused by $Ar/O_2$ plasma then the $N_2$ plasma treated or untreated film [90].

Cold plasma treatment enhances adhesion between different surfaces and enables the production of multilayer films as shown in Figure 4. Honarvar, Farhoodi [43] reported that the application of atmospheric cold plasma improved the attachment of carboxymethyl cellulose with polypropylene films. Similarly, Fazeli, Florez [57] found a significant en-hancement of adhesion between air plasma-treated cellulose fiber and thermoplastic starch. Several other studies have shown that plasma treatment enhanced the attachment between different polymer layers like polyethylene terephthalate/polypropylene films assembled with chitosan loaded with different preservatives [44], low-density polyethylene coated with carboxymethyl cellulose (CMC), or collagen containing cinnamaldehyde [45].

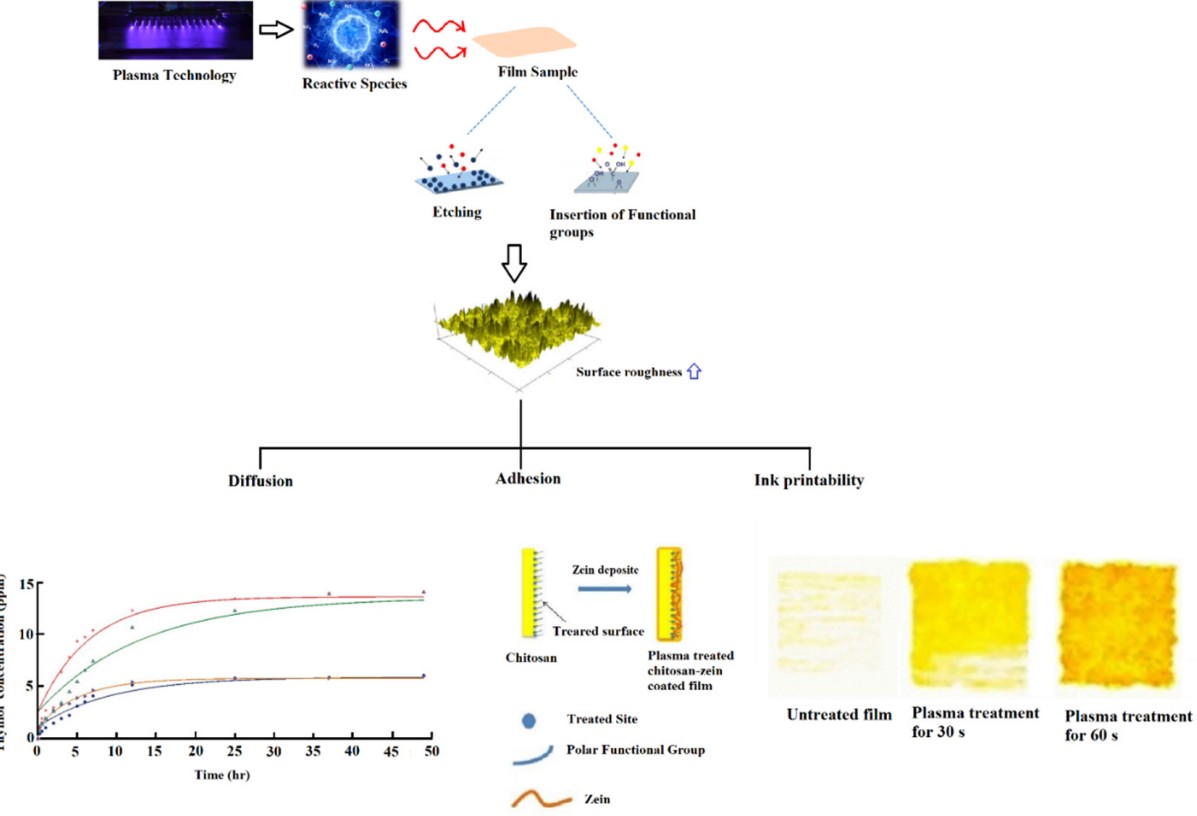

**Figure 4.** The primary effect of cold plasma treatment on the surface biopolymer. Reprinted with permission from Refs. [81,83], Elsevier, 2022. Reprinted from Refs. [88,91].

The increased roughness enhances the ink printability of the film as shown in Figure 4. The plasma-activated surface of non-absorbent materials like PLA, enhances ink adhesion influencing printability to the substrate. Izdebska-Podsiadły [92] reported that cold plasma involving $O_2$ and Ar modified the printability of PLA film. Similarly, the printability of PLA film, and starch/poly($\varepsilon$-caprolactone) films improved when treated with cold plasma involving $O_2$ and air plasma, respectively [79,93].

### 3.2. Contact Angle

Contact angle indicates the tendency of a liquid drop to spread out and adhere to the surface of materials, reflecting its wettability and hydrophilicity [58,79]. The higher the contact angle values, the higher the tendency of the surface to repel water. In general, the application of cold plasma onto film surfaces decreases the contact angle values indicating an increase in the hydrophilicity and the wettability of the films. The effect of plasma treatment on the contact angle of the different biopolymers is summarized in Table 2. Application of cold plasma onto the film surface generates reactive species and free radicals resulting in the amplification of several polar groups like -COOH, -OH, and CO leading to increased polarity, surface tension, and surface free energy, eventually increasing the hydrophilicity and wettability of the film surface [52,80,93]. Another probable reason is the surface roughness caused by the etching effect that aids in the spreading of liquid onto the surface [70,80,94]. As reported in the published literature, the water contact angle is the function of plasma treatment time and voltage. Plasma treatment at higher voltages and with longer exposure times increases the surface roughness and polar groups, resulting in hydrophobicity and wettability of the film surface. In contrast to the general trend where the application of cold plasma decreases contact angle, DBD plasma-treated starch film showed an increase in the contact angle values. This could be due to the oxidation of the hydroxyl group into carbonyl groups, resulting in the formation of the new hydrogen bonds. These hydrogen bonds may have decreased the availability of polar groups on the surface, leading to the enhancement in hydrophobicity of the film surface [65]. Similarly, contact angle values increased for corn starch thermoplastic films treated with $SF_6$ plasma [95]. Overall, it can be concluded that the contact angle is a factor of plasma gas composition, as well as voltage and exposure time. The contact angle of a film is an important criterion to decide its suitability in packaging applications [96,97]. The wettability of a film affects the coating, printing, absorbance, adhesion, and frictional properties of the film surface [97,98].

### 3.3. Molecular Properties of the Film
Fourier Transform Infrared (FTIR) Study

A microstructural study of the biopolymer-based film provides insight into the molecular interactions of the different components present in the film network. The understanding of molecular interactions is important as it influences the physical, thermal, and mechanical properties of the film. Cold plasma treatment is a novel technique used to enhance the structural features of the biopolymer-based film. One of the most prominent methods to investigate the structural changes in the biopolymer film is Fourier transform infrared (FTIR). It has been reported that plasma treatment induces a number of chemical reactions like molecular rearrangements, dehydration, and the hydrogenation of molecules [58].

The application of plasma creates a reactive site on the biopolymer film surface. Sheikhi, Hosseini [99] treated starch films with plasma (air and $O_2$), and Moosavi, Khani [100] treated gluten and whey protein with cold plasma. The FTIR study showed that bands near 2940 cm$^{-1}$ and 2925 cm$^{-1}$ decreased and this was attributed to the abstraction of protons from the carbon atom of C-H of starch and proteins resulting in the creation of reactive sites on the biopolymers [43,99,100]. The studies have shown that the starch-based film treated with cold plasma incorporates oxygenated functional groups into the film matrix resulting in an increase in hydrogen bonding and crosslinking. Arolkar, Salgo [79] investigated the structural modification of corn starch/poly-$\varepsilon$-caprolactone films using FTIR and reported that the application of air plasma increased the intensity of peaks at

approximately 3600–3000, 1265, and 1016 cm$^{-1}$ suggesting the increase in the oxygen-containing functional groups like O-H and C-O (Figure 5I). This increase in the peak intensity could be due to the interaction between oxygen-containing species in the plasma and the substrate [79]. A similar result was reported for corn starch film when treated with DBD where the increase in absorption bands at around 1016 cm$^{-1}$ (C-O-C), 1154, and 1081 cm$^{-1}$ (C-O-H) indicated an increase in C-O groups. A shift in the absorption bands at about 1648 and 1580 cm$^{-1}$ (δ (O-H)) indicated an increase in hydrogen bonding and these phenomena could be attributed to surface oxygenation [58,82,99]. Further, Sifuentes-Nieves, Hernández-Hernández [101] and Sifuentes-Nieves, Mendez-Montealvo [85] studied the impact of cold plasma on starch films with low (30%) to high (50–70%) amylose content and reported that the application of different types of plasma (such as Hexamethyldisiloxane cold plasma, DBD, and radiofrequency plasma) altered the short-range crystalline structure of starch. This was indicated by a band ratio 1047/1022, where a peak at 1047 cm$^{-1}$ denoted an ordered or crystalline structure and at 1022 cm$^{-1}$ represented the amorphous region of starch [102,103]. As shown in Figure 5II, after the plasma treatment the intensity at 1022 cm$^{-1}$ decreased and 1047/1022 increased for all of the films. This indicates that the etching phenomena allowed the plasma reactive species to penetrate and reorder the amylopectin chains, thereby creating a highly ordered network in the starch film matrix [85,101]. A study by Sifuentes-Nieves, Hernández-Hernández [101] further reported that the peak at 3300 cm$^{-1}$ corresponding to the O-H group disappeared in all the plasma-treated films and this was attributed to the impact of plasma reactive species altering the water retention capability of the starch film by the formation of a coating on the film surface [101]. Similarly, Goiana, de Brito [58] also reported that DBD plasma treatment of corn starch for 20 min demonstrated a reduction in the intensity of the peak in the range of 3500–3000 cm$^{-1}$ at approximately 3295 cm$^{-1}$, implying the possible dehydration of film during the plasma treatment and thereby reduced hydrophilicity [58].

Several studies have also reported the modification of protein structure plasma treatment. Similar to starch, the application of cold plasma increased oxygenated groups (C-O and C-O-C) in gluten and whey protein and this was evidenced from the increase in peaks at 1105 cm$^{-1}$ and 1048, & 1258 cm$^{-1}$, respectively [100]. The characteristic peak in the range of 3700–3000 cm$^{-1}$ can be attributed to the N-H and O-H stretching of proteins. Atmospheric cold plasma treatment of zein film showed a shift and slight increase in peak intensity 3300 cm$^{-1}$ attributable to increased polar groups on the film surface [80]. Such modifications were also observed from cold plasma treatments of 60 s in composite films like zein/chitosan [94], and zein/PLA [46] where the shift and increase in peak intensity in the region from 3700–3000 cm$^{-1}$ indicates the formation of a hydrogen bond between the zein-PLA and zein-chitosan interface. In these studies, it was further reported that the plasma treatment for the 60 s in zein/PLA composite film and 120 s in zein/chitosan treatment showed an extra peak in the carbonyl region 1800–1650 (at 1754 cm$^{-1}$) which could be ascribed to the protein oxidation or peptide bond breakage [46,94]. Chen, Dong [94] also reported that the variation of the peak intensity in the region of 1300–750 cm$^{-1}$ (at 1097 cm$^{-1}$) could also be attributed to the interaction between C-O (chitosan) and amide (zein) groups.

In the protein structure the FTIR spectrum in the range of 1700–1600 cm$^{-1}$ (amide I) corresponds to the secondary structure components (1659–1660 cm$^{-1}$: α-helix; 1660–1670 cm$^{-1}$: β-turn; 1610–1640 cm$^{-1}$: β-sheet; and 1640–1650 cm$^{-1}$: random coil) and 1600–1500 cm$^{-1}$ (amide II) refers to N-H bending and C-N stretching modes indicates the environment for hydrogen bonding [80,94,104]. It has been reported that the application of cold plasma transformed or rearranged the secondary structure components into a β-sheet structure.

Jahromi, Niakousari [104] investigated the impact of DBD atmospheric cold plasma on sodium caseinate film and reported that the peak intensity in the region of 1637–1645 cm$^{-1}$ decreased with the increase in plasma treatment and the peak disappeared in the film treated for 10 min, indicating the decrease in random coil content. However the peak intensity at 1623 cm$^{-1}$ and 1689 cm$^{-1}$ increased for the treatment time of 2.5 to 5 min signifying the increase in β-sheet structure [104]. Dong, Guo [80] investigated the impact of

atmospheric cold plasma on zein film and reported that with an increase in treatment time (0 to 60 s) β-sheet and random coil content increased by 25.75% and 16.37%, respectively. On the contrary, there was a decline in the β-turn and α-helix by 13.45% and 20.63%, respectively, indicating the transformation of α-helix and β-turn into β-sheet and random coil. Similarly, Chen, Dong [94] reported the transformation of β-turn and random coil into β-sheet and α-helix in zein/chitosan film after the application of atmospheric cold plasma for 0–120 s. Wu, Liu [105] studied the impact of voltage and treatment time of DBD plasma on casein edible films. It was reported that at a constant treatment time of 60 s with the increase in the voltage from 0 to 70 V, α-helix and random coil content decreased by 53.78% and 46.83%, respectively, in comparison to control film. Whereas the β-sheet content increased by 67.51%. Similarly, β-turn content increased by 20.95% up to a voltage of 60 V and then decreased by 20.95% when the voltage was increased to 70 V. Such a result indicates the transformation of α-helix into β-sheet. It was further investigated at a constant 50 V power, and the α-helix and random coil content decreased with the increase in treatment time up to 90 s and then again increased, although the overall content was less than the control film. Similarly, β-sheet increased with the increase in treatment time up to 90 s and then decreased, whereas the increase in β-turn content was quadratic. Such variation in the secondary structure of the protein can be ascribed to the destruction of orderly structure at short treatment times and the reproduction of orderly structure at higher treatment times [105]. Also, as shown in Figure 5III, an increase in the peak intensity at 1543 cm$^{-1}$ (zein film) and 1533 cm$^{-1}$ (zein/PLA), corresponding to amid II, indicates the impact of plasma on hydrogen bonding resulting in the modification of secondary structures [46,80]. Such enhancement in β-sheet structure was also reported for Ar and air (10 min) treated protein films (whey protein and gluten) and was evident from the decrease in peak intensity at 1234–1239 cm$^{-1}$ [100].

Overall, it was observed that the application of plasma influences the structure of the biopolymer-based film by various mechanisms like creating a reactive site, crosslinking, reordering, and enhancing the structural arrangement of the polymeric network. These structural modifications affect the various properties like mechanical, thermal, and barrier properties of the biopolymer films.

### 3.4. Mechanical Properties

Mechanical properties such as tensile strength (TS) and elongation at break (EAB) of the films are key parameters in packaging applications to preserve food quality and integrity during handling, storage, and transportation [43,52,94]. The application of plasma to film induces physicochemical phenomena like etching, degradation, and cross-linking altering the mechanical properties of the films [52,95]. The bombardment of plasma (ionic species) on the film surface cleaves C-C and C-H bonds and generates free radicals that create linkage with the surface radicals or participate in chain reactions resulting in the significant improvement in mechanical properties of the films [100,106]. Dong, Guo [106] reported that atmospheric cold plasma treatment for 45 s significantly increased the TS and EAB of a zein film. However, increasing the treatment time to 60 s was ineffective in improving the mechanical properties. The increase in TS and EAB of the zein film could be due to the cross-linking that occurred during plasma treatment. The plasma discharge could have oxidized oxygen into an ozone that transformed the free S-H group to S-S. Cross-linking by strong intermolecular and intramolecular S-S in the zein matrix may have resulted in the higher TS and EAB. Such findings for zein/chitosan film, zein, and zein/polylactic acid films were reported in the literature [46,80].

Goiana, de Brito [58] treated the corn starch-based film with DBD plasma and reported that with the increase in treatment time from 0 to 20 min, TS increased significantly. Similarly, Sheikhi, Mirmoghtadaie [95] also demonstrated that TS of Ar plasma treated starch film increased significantly with the increase in treatment time from 0 to 12 min [95]. The increase in TS could be attributed to the probable introduction of cross-links like carbonyl groups after plasma treatment that promotes hydrogen bonding resulting in the enhanced

structural integrity of the film matrix [58,95]. They also reported that improvements in TS could be due to the etching effect, which reduces some of the defects on the surface of the film that cause premature failure [95]. These studies showed that TS of the biopolymer films increased with an increase in plasma exposure time due to the longer chemical interaction between the plasma-generated radicals and the polymer surface.

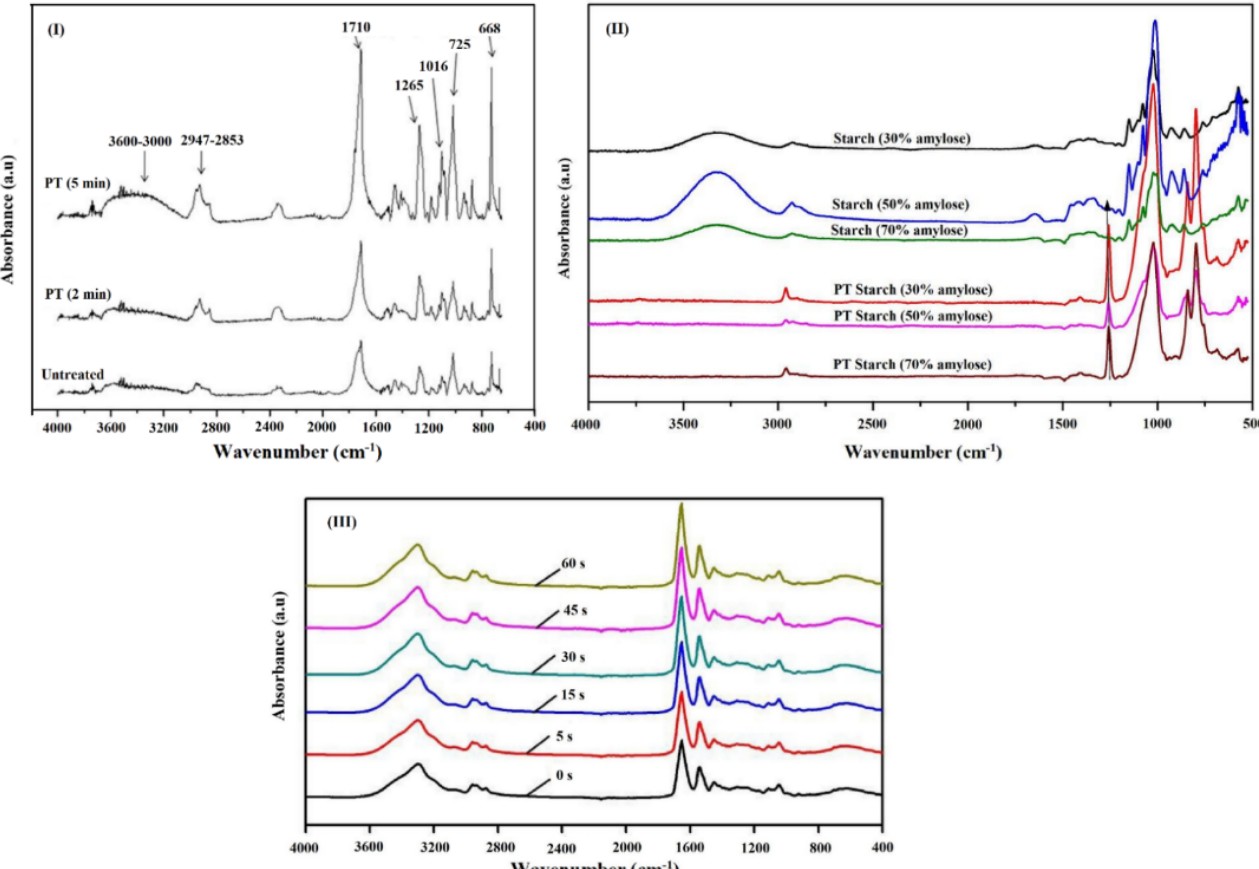

**Figure 5.** ATR-FTIR spectra: (**I**) air plasma treated corn starch/poly-ε-caprolactone films. Reprinted with permission from Ref. [81], Elsevier, 2022; (**II**) hexamethyldisiloxane (HMDSO) plasma-treated starch film. Reprinted with permission from Ref. [81], Elsevier, 2022; (**III**) atmospheric cold plasma treated zein films. Reprinted with permission from Ref. [81], Elsevier, 2022.

Dong, Li [49] studied the impact of applied voltage and plasma treatment time on the mechanical properties of polycaprolactone (PCL) grafted zein film. In general, TS increased with the increase in voltage and time. In particular, the plasma treatment for more than 15 s significantly enhanced the TS and EAB due to the intense chemical interaction between zein and PCL. However the application of a higher voltage of 100 V decreased the TS of the composite film. They also reported that the overall enhancement in EAB due to plasma treatment was negligible when compared to PCL film. This suggests that mechanical property is largely reliant on the bulk structure of the film rather than the surface layer [49].

Besides treatment time and voltage, the types of gases used to generate plasma reactive species also have an impact by altering the mechanical properties of the packaging film. Moosavi, Khani [100] reported that the application of air and argon plasma for 5 and 10 min in whey protein film increased the TS significantly. Increasing the treatment time up to 15 min significantly increased the TS for air plasma treated film, however a decline was observed for argon plasma treated film. This could be due fact that the ionization yield of Ar plasma is higher than air plasma. Furthermore, Ar is more effective in crosslinking, with the formation of free radicals and double bonds presenting a higher impact on the mechanical properties of the films. During the formation of air plasma, a portion of electrical energy

is used in the vibrational and rotational excitation of the gas molecule [52,100]. Similarly, Chen, Dong [94] modified zein and chitosan-based composite films through the application of cold plasma for 60 s and found that the TS and EAB improved significantly compared to zein or composite of zein/chitosan without plasma treatment. However high energy in the plasma and the longer treatment of film for 90 s decreased the compactness of the zein/chitosan composite, which resulted in a drop in TS and EAB. Ledari, Milani [78] reported that the application of cold plasma using $O_2$, $N_2$, air, Ar, and ethanol-argon gases alone cannot change the mechanical properties of gelatin films significantly.

Overall, it was observed that increases or decreases in the mechanical properties of biopolymer-based films treated with cold plasma are a function of various factors including the type of polymer (protein or polysaccharide), types of gases used to generate plasma reactive species, processing time, and voltage.

### 3.5. Thermal Properties

Differential scanning calorimetry (DSC) and thermogravimetric analysis (TGA) are the two most important approaches in understanding the thermal properties of biopolymers and their films. Thermal studies not only provide information on the changes in heating/cooling behaviors but also scrutinize the thermodynamic and thermophysical properties [52]. TGA shows the thermal stability of the samples and the DSC techniques determine the melting temperature ($T_m$), glass transition temperature ($T_g$), denaturation temperature ($T_d$), and enthalpy ($\Delta H$). The $T_g$ is the temperature at which the polymer chains become more mobile, meaning that they have more freedom of movement, whereas the melting temperature is the temperature at which crystalline structures are lost and the polymer chains become a disordered liquid [107]. Thermal parameters are important in determining the processing conditions of the biopolymers and their applications in packaging, as heat sealing is a critical point in the packaging [58,108]. Table 2 summarizes the thermal properties of biopolymer films exposed to plasma treatment.

Pankaj, Wan [87] scrutinized the impact of high voltage (80 kV) atmospheric cold plasma treatment (HVACP) on thermal properties of (corn, potato, rice, and tapioca) starch films. It was reported that HVACP treatment created a highly ordered structure in the films and thus showed higher $T_g$ and $\Delta H_m$ as compared to control films. It was also observed that the $T_g$ and $\Delta H_m$ values were highest for corn starch with high amylose content whereas it was found to be lowest for the rice starch with the lowest amylose content [87]. In another study, Dong, Guo [80] studied the thermal properties of zein films and DSC results showed that atmospheric cold plasma treatment for 60 s enhanced the denaturation temperature ($T_{den}$), which could be attributed to the crosslinking of the polar compounds generated at the film surface. A similar result was reported for zein-polylactic acid film, where plasma treatment for the 60 s and 120 s led to the formation of more thermally stable films and was attributed to the enhancement of intermolecular hydrogen bonding interactions between zein and PLA molecules with the increase in plasma treatment [46]. They also reported that the TGA of the zein films reflected that the thermal degradation temperature ($T_{deg}$) increased for plasma-treated film and was found to be highest for the film treated for 45 s. Similarly, TGA of the zein-polylactic acid composite film showed that plasma-treated film (for the 60 s and 120 s) demonstrated a lower thermal degradation (73.66%) as compared to untreated composite film (77.31%). This increase in thermal stability may be attributed to the enhancement of compatibility between two different phases upon the application of cold plasma [46].

Apart from the plasma treatment time and voltage, types of plasma forming gases are also important factors that control the thermal stability of the biopolymer-based films. Ledari, Milani [78] carried out DSC of gelatin-corn oil emulsion-based films treated with $O_2$, $N_2$, air, Ar, and ethanol-argon (EtOH-Ar) plasma. The DSC analysis of these films showed that plasm treatment using different plasma forming gases had a different impact on the thermal properties like melting temperature ($T_m$) and melting enthalpy ($\Delta H_m$). On the application of air, $T_m$ and $\Delta H_m$ increased significantly from $51.9 \pm 0.1$ to $58.9 \pm 0.1$ (°C),

and $65.5 \pm 0.04$ to $123.8 \pm 0.1$(J/g), respectively, and this can be attributed to changes in the chain mobility of the films. The etching effect and the loss of moisture from the film surface result in the modification of $T_m$ and $\Delta H_m$ as the presence of water can act plasticizer and increase the chain mobility. Similarly, the application of other gases like $O_2$, $N_2$, and EtOH-Ar increased the $T_m$ and $\Delta H_m$ with the exception of Ar gas, which had no significant impact on the thermal properties of the films [78]. In another study, Oh, Roh [109] investigated the impact of cold plasma treatment with different gases ($O_2$, $N_2$, air, He, and Ar) on the edible film prepared from defatted soybean meal. It was reported that the application of He and Ar increased the $T_g$ compared to the untreated film, which may have been due to the introduction of radicals at the polymer surface resulting in the formation of a cross-linked network and leading to the increase in $T_g$. On the other hand, $T_g$ decreased in films after $O_2$ cold plasma treatment due to the amalgamation of oxygen-containing functional groups at the biopolymer surface, enhancing the surface molecular mobility and resulting in the observed decrease in $T_g$ values [109].

A few studies reported that plasma treatment had no impact on the thermal properties of biopolymer-based film from PLA [110], nisin-coated PLA [48], and zein [75]. Other studies reported the negative impact of cold plasma treatment on the thermal properties of biopolymer-based films. Pankaj, Bueno-Ferrer [84] studied the effect of dielectric barrier discharge (DBD) plasma treatment on sodium caseinate film and showed that $T_g$ of the plasma-treated film decreased as compared to the untreated film. The decrease in $T_g$ can be ascribed to chemical etching due to the breakdown of the chemical bonds, chain scission, or chemical degradation of the plasma-treated biopolymer. Similarly, Romani, Olsen [107] studied the impact of glow discharge plasma on fish protein films. The DSC result showed that the application of plasma for 2 and 5 min decreased $T_g$, $T_m$, and $\Delta H_m$ of the film significantly as compared to the untreated film. The changes in these thermal parameters suggest that the biopolymer matrix is affected.

*3.6. Water Barrier Properties*

Water vapour permeability (WVP) of packaging material is an important parameter in determining the moisture barrier property of the packaging materials that affect the resulting shelf life and quality of the packaged food. The poor moisture barrier property of biopolymer-based materials is one of the major shortcomings in packaging applications [78,111]. Different modification techniques are adopted to enhance the barrier property or at least avoid any drastic change in the permeability in order to augment the usage of biopolymers in food packaging applications.

From the literature, as summarized in Table 2, the application of plasma onto film surfaces shows a range of possible effects. Some authors reported that plasma treatment decreased the WVP, while others observed that plasma treatment had no significant impact on the barrier property or in some cases increased the WVP. Cold plasma treatment decreased the WVP of starch/PCL and starch/PLA composite films by approximately 94% [112]. Due to the enhancement in adhesion properties caused by plasma treatment, different composite films like zein/chitosan [94], zein/PLA [46], and whey protein concentrate/wheat cross-linked starch [113] films also showed a significant decrease in WVP. Dong, Guo [80] treated zein film using atmospheric cold plasma and reported that WVP decreased by approximately 24% as compared to control films. In general, plasma treatment of the film creates surface roughness and increases the tortuosity of the diffusion pathway of water resulting in the decrease in WVP [52,94]. On the contrary, Arolkar, Salgo [79] reported that the WVTR increased when treated with plasma for two minutes or beyond due to the etching effect that facilitates water vapor transport.

Studies demonstrating that no impact is caused by plasma on barrier properties include the work of Ledari, Milani [78]. They showed that plasma treatment with different gases like $O_2$, $N_2$, air, Ar, and ethanol-argon (EtOH-Ar) had no significant effect on the WVP of gelatin films. Similarly, Sheikhi, Hosseini [99] found no significant change in the starch film when treated with air and $O_2$ for 4, 8, and 12 min. Although the application

of different gases increased the polar functional groups (like peroxide, carboxylic acid, and hydroxyl, C-N, C=N, and amide bonds) at the surface of the polymer and induced cross-linking resulting in higher cohesive energy density, it did not affect the WVP. This indicates that WVP is not only dependent on the surface properties but also depend on thermodynamic properties, vapour pressure, and concentration gradient across the film surface depending on diffusion and the solubility mechanism [52,78,83,99].

### 3.7. Oxygen Permeability (OP)

OP is another key parameter of packaging materials that determines the oxygen transport across the film to enhance shelf life and quality of the packaged food. The presence of oxygen modulates the development of the different reactions like oxidation involving components responsible for the color and aroma of the food product [78,100]. When packaging oxygen-sensitive products, the materials with the least amount of oxygen permeability are preferred. Biopolymer-based packaging materials have lower oxygen barrier properties when compared to conventional plastic packaging. Therefore, some modifications of biopolymers are necessary to restrict oxygen permeability to enhance its application in the food packaging industry. Table 2 summarizes the effect of cold plasma on the OP of the biopolymer-based films.

The literature reports that biopolymer films treated with cold plasma were observed to have increased oxygen barrier properties or that the treatment did not alter the inherent properties of biopolymers. Ledari, Milani [78] treated gelatin film with $O_2$, $N_2$, air, Ar, and ethanol-argon (EtOH-Ar) and found that OP decreased significantly. Similarly, whey protein film and gluten film showed a significant drop in the OP when treated with air and argon (Ar) plasma [100]. Sheikhi, Hosseini [99] treated starch film at low pressure with air and $O_2$ plasma and reported that OP decreased significantly.

In general, the decrease could be attributed to the enhanced cross-linking between polymer chains resulting in a compactness that reduces free volume and prevents oxygen diffusion. Several other studies also suggest that plasma exposure induces physiochemical changes in the polymeric surface, like a breakdown of the C-C and C-H bond, creating free radicals. The interaction between the generated free radicals and activated plasma species like excited ions, electrons, and molecules leads to the development of intermolecular and interchain bonds and thereby enhances the oxygen barrier property. Moreover, OP depends upon polymer crosslinking and polarity. The application of plasma (like air and Ar) enhances the hydrophilic groups on the surface of the film increasing the polarity leading to the increase in cohesive energy density making it more difficult for permeants to open the polymer chains and permeate through its matrix [78,99,100].

Some authors have reported that plasma treatment had no significant effect on the oxygen permeability (OP) of the film. Pankaj, Bueno-Ferrer [83] treated gelatin films with dielectric barrier discharge (DBD) atmospheric air cold plasma, resulting in no significant changes in the OP. Similarly, chitosan film treated with dielectric barrier discharge atmospheric air cold plasma showed no significant change in OP [86]. The authors reported that although the DBD atmospheric air cold plasma affects the surface properties of the film, bulk properties were unaltered. This may be the reason that no significant change in the OP of the DBD atmospheric air plasma treated films was observed. It is important to emphasise that gas permeability follows a combined mechanism of diffusion and solubility where the permeant (gas) passes through the void present between the polymer chains in the film network [83].

### 3.8. Antimicrobial Properties

A novel cold plasma technique is applied to modify polymer surface attributes, the decontamination of packaging surfaces and food processing instruments, and to enhance the safety level of several food products [52,114]. The active packaging films loaded with bioactive compounds like essential oils, peptides, or functional components when exposed to cold plasma show higher antimicrobial activity.

Loke, Chang [45] prepared active packaging film from cold plasma treated low-density polyethylene (LDPE) coated with carboxymethyl cellulose (CMC) or collagen (COL) containing cinnamaldehyde (CMAL). The application of cold plasma enhanced the attachments between the LDPE and CMC/CMAL or COL/CMAL and showed strong antibacterial activity against *S. aureus* and *E. coli.* The composite film LDPE/COL/CMAL demonstrated a strong antimicrobial effect against both these bacteria when compared to LDPE/CMC/CMAL due to the COL's higher retention capability. Further, the LDPE/COL films containing 0, 2.0, 4.0, 6.0, and 8.0% CMAL were used for packaging tilapia fillets and stored for 14 days. The total plate count (TPC) in the fish decreased with the increase in the CMAL concentration and the LDPE/COL film containing 6.0 and 8.0% CMAL enhanced the shelf life of fillet by at least three days. Also, the antimicrobial efficacy of the prepared LDPE/COL/CMAL film was tested against *V. parahaemolyticus* in the packaged tilapia fillets stored at refrigerated temperatures (4 °C) and reported that the bacterial population at the end of the storage period was found to decrease by 1.27–1.92 log CFU/g for the film loaded with 6.0 and 8.0% CMAL as compared to the control film. The reduction in the microbial count can be attributed to the antimicrobial effect of CMAL during the storage period [45]. Similarly, plasma-treated polypropylene (PP) films coated with carboxymethyl cellulose (CMC) loaded with Zataria multiflora essential oil (ZEO) and showed the antimicrobial activity against Gram-positive (*S. aureus*) and Gram-negative (*E.coli, S. typhimurium*, and *P. aeruginosa*) bacteria [43]. As shown in Figure 6I the zone of inhibition increased with the increase in the concentration of ZEO. The antimicrobial activity of the PP/CMC/ZEO is higher when compared to the CMC/ZEO film [115] and this can be attributed to the PP layer that acts as a barrier against the loss of ZEO. The attachment between PP and CMC/ZEO is facilitated by plasma treatment [43]. Such antimicrobial activity on *S. aureus*, *B. subtilis*, and *E. coli* was also reported for polyethylene terephthalate (PET)/polypropylene (PP) film assembled with chitosan loaded with sodium benzoate (1.0–5.0%), potassium sorbate (1.0–5.0%), calcium propionate (1.0–5.0%) and ε-poly-lysine (0.2–1.0%). All the plasma assembled films showed strong antimicrobial activity [44]. Although cold plasma treatment had no direct impact on the enhancement of antimicrobial activity, it increases adhesion between different layers and thereby protects antimicrobial components from loss resulting in higher antimicrobial activity.

It is observed that plasma treatment facilitates the coating of functional components on the polymer layer and thereby increases antimicrobial efficacy. For instance, Wong, Hou [116] prepared gallic acid (GA) coated polyethylene (PE) film applying plasma treatment (30 W for 60 s). The PE/GA active film reduced the growth of *E.coli* and *S. aureus* by 0.5–1.1 log reduction at a concentration above 1.0%. Such antimicrobial activity was also reported for plasma-treated polylactic acid (PLA) film coated with nisin where the active film showed a log reduction of 3.23 against *Listeria monocytogene*, whereas pristine PLA film could not inhibit its growth. It was observed that with the increase in plasma treatment time (0–60 s) the microbial reduction increased and this could be ascribed to the content of nisin absorbed on the surface of the PLA. This indicates that cold plasma treatment influences the absorption capacity of PLA film and thus the content of nisin thereby affecting the antimicrobial activity of the film [48].

The diffusion rate of antimicrobial compounds from the plasma-treated polymer matrices and its release rate can be a critical factor influencing the antimicrobial activity of the active films. De Vietro, Conte [117] applied low-pressure aerosol-assisted plasma to prepare polycarbonate film coated with copper (organic-inorganic) and investigated the antimicrobial activity against (*P. fluorescens* and *P. putida*). The authors reported a three-log reduction (from 108 to 105 CFU/mL) compared to the control polycarbonate film. The reduction in the microbial count was attributed to the release of $Cu^{2+}$ ions by coatings that penetrate the microbial cell resulting in their disruption [117]. Such results are consistent with other studies where Pankaj, Bueno-Ferrer [88] showed that DBD plasma treatment increased the diffusion rate of antimicrobial thymol from zein matrix. This was ascribed to the etching effect that led to an increase in roughness and resulted in a

decrease in the thickness [88]. Karam, Jama [90] and Karam, Casetta [89] showed that plasma treatment enhanced the release of nisin from LDPE/nisin film and thus increased the microbial inhibition against *L. innocua* as shown in Figure 6II.

Some studies demonstrated that plasma treatment had no significant effect on the inhibition of microbial growth. Chen, Ali [81] showed that cold plasma had no significant effect on chitosan/ciprofloxacin hydrochloride antimicrobial film and further coating with zein had drastically reduced the release of ciprofloxacin hydrochloride resulting in reducing the antimicrobial effect. This could be attributed to the fact that plasma treatment may have enhanced intermolecular interaction (crosslinking) leading to firm adhesion between chitosan and zein, resulting in the increase in a barrier for the diffusion of ciprofloxacin hydrochloride [81]. Such reduction in the release of the antimicrobial compound due to cross-linking was also reported for sodium tripolyphosphate crossed linked chitosan film loaded with ciprofloxacin [118].

Overall, the impact of plasma treatment on the antimicrobial activity of the film containing functional compounds is critical. The strategic combination of polymers with antimicrobial substances treated with cold plasma can be a potential approach when preparing antimicrobial packaging materials suitable for food packaging applications.

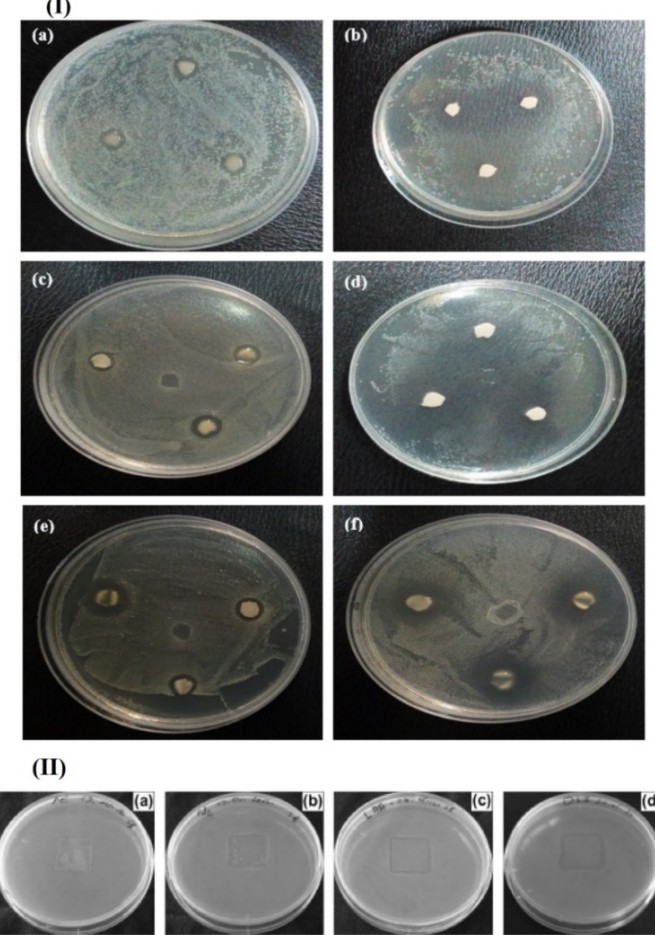

**Figure 6.** (**I**) Antimicrobial activity of cold plasma treated PP/CMC/ZEO film against *S. aureus*: (**a**) PP/CMC/1.0% ZEO: 9.24 ± 0.1 mm; (**b**) PP/CMC/4.0% ZEO: 30.26 ± 0.4 mm; against *E. coli*; (**c**) PP/CMC/1.0% ZEO: 8.71 ± 0.0 mm; (**d**) PP/CMC/4.0% ZEO: 26.73 ± 0.1 mm; against *S. ty-phimurium*; (**e**) PP/CMC/1.0% ZEO: 8.82 ± 0.2 mm; (**f**) PP/CMC/4.0% ZEO: 24.94 ± 0.1 mm. Reprinted with permission from Ref. [81], Elsevier, 2022; (**II**) antibacterial activity of plasma treated LDPE against *L. innocua*: (**a**) LDPE/nisin; (**b**) $N_2$ plasma treated LDPE/nisin; (**c**) acrylic acid treated LDPE/nisin; and (**d**) $Ar/O_2$ plasma treated LDPE/nisin. Reprinted with permission from Ref. [81], Elsevier, 2022.

### 3.9. Biodegradability

Biodegradable materials are degraded by the enzymatic action of living microorganisms such as bacteria, yeast, and fungi. Biodegradation is investigated in various environments such as soil burial, landfill and compost simulations, and microorganisms [79,109,119]. The impact of cold plasma treatment in the biodegradation of polymers has been reported in the literature as summarized in Table 2. Arolkar, Salgo [79] investigated the degradation of corn starch/poly(ε-caprolactone) films following soil burial method and the extent of degradation was reported in terms of an alteration of TS and EAB. It was observed that the rate of degradation was higher for air plasma treated films as compared to untreated film and it increased with increase plasma treatment time. In another study, Chen, Chen [46] reported that the biodegradability of the plasma-treated zein-PLA composite film was higher than that of untreated films. The higher degradation of the plasma-treated film was attributed to the increase in the surface area of the porous structure that enhanced the accessibility to the microbes present in the compost into the film resulting in the higher degradation. A similar result was reported for plasma-treated defatted soybean meal film, where the plasma treatment increased the surface area of the film by increasing the roughness of the film, facilitating microorganisms in the compost in faster degradation [109]. In another study, Song, Oh [93] investigated the biodegradation of PLA sachets for a period of 0 to 35 days and reported that at the end of 28 days, the extent of degradation of plasma-treated PLA sachets was higher than in untreated sachets. At the end of the 35 day period, the treated PLA sachets were completely degraded whereas the untreated sachets remained as shown in Figure 7 [93]. Overall, it was observed that plasma treatment modifies the film surface and enhances the rate of microbial degradation.

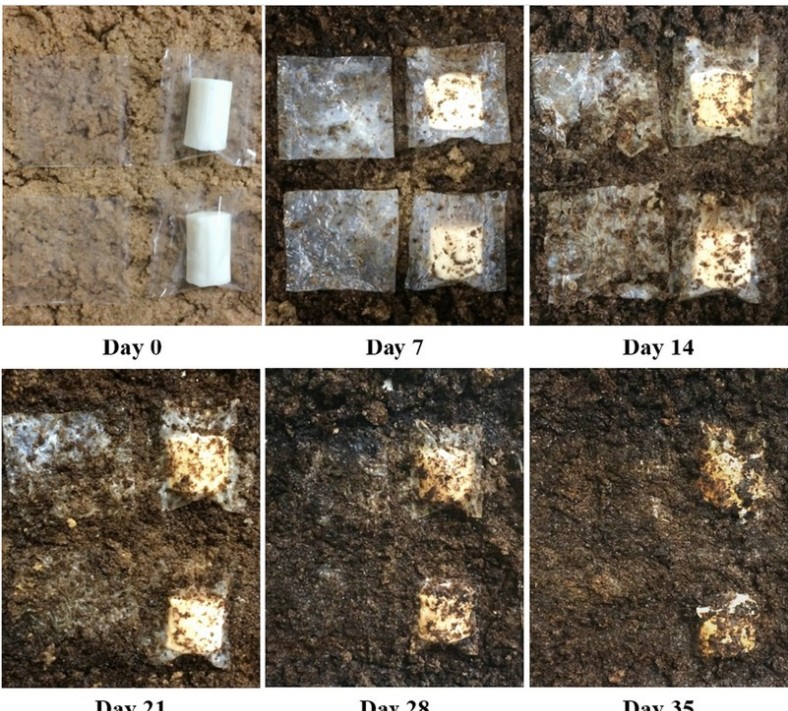

**Figure 7.** Effect of cold plasma treatment on biodegradability of PLA film when composted at 90% RH and 37 °C for 35 days. On each day, untreated and treated samples are kept in upper and lower positions, respectively. Reprinted with permission from Ref. [93], Wiley Online Library, 2022.

## 4. Safety Concerns Relating to the Application of Cold Plasma for the Modification of Food Packaging Films

The application of plasma in food processing/packaging generates reactive species that come in direct or indirect contact with food surfaces. The interaction between plasma

reactive species and the matrices of food or biopolymers is complicated and limited studies have been carried out investigating the potentially detrimental effects to human or animal health [52,120]. Chen, Lin [121] reported that He gas-plasma treatment of di-ionized water for 30 min generated ROS (reactive oxygen species) and RNS (reactive nitrogen species) caused significant apoptosis [121]. In another study, Heslin, Boehm [122] assessed the cytotoxic and mutagenic potential of DBD plasma-treated iceberg lettuce in an in vitro CHO-K1 (mammalian cell) model and short-term toxic effects in an in vivo *Galleria mellonella* larva model. The results showed a low in vitro cytotoxic effect and spontaneous mutations, however reported a strong in vivo toxicity with less than a 10% larva survival when injected with lettuce broth treated for 5 min [122]. Few studies have been found to evaluate the toxicology of plasm-treated food constituents. Some studies have reported that the plasma reactive species induces alterations in food constituents, like modification of amino acids in proteins, oxidation of high molecular weight compounds to organic acids, and peroxidation of lipids resulting in undesirable metabolites like short-chain aldehydes, keto-acids, hydroxyl acids, and short-chain fatty acids [51,120,123]. Other effects of the application of cold plasma on food commodities include a reduction in firmness of fruits and vegetables, enhanced discolouration, and surges in acidity content [52].

Some studies showed that the application of cold plasma in food products caused no acute toxicity, although the work conducted on this has been limited. For instance, Kim, Sung [124] evaluated the mutagenicity and immune toxicity of sausage prepared with plasma-treated water as a nitrite source and reported that mutagenicity and inflammatory response was negative in mice fed with the plasma-treated sausage [124]. Similarly, Jo, Lee [125] reported that atmospheric plasma-treated winter mushroom powder caused no mutagenicity or acute toxicity in rats fed with 5000 mg/kg body weight [125]. A few studies have been found to investigate the toxicity of biopolymer-based films treated with plasma. In one of these studies, Han, Suh [126] investigated the safety of plasma-treated soya film by determining the acute and subacute oral toxicity in a rat model. It was reported that the rat fed with 5000 mg/kg body weight (Single-dose acute) or subacute 1000 mg/kg body weight/day for 14 days showed no acute toxicity resulting in the death of the rats. However, a change in blood components (like hematocrit, hemoglobin, bilirubin, creatinine, and aspartate aminotransferase) was observed. These changes were irrelevant to toxicity as their level were within acceptable physiological ranges [126].

Overall, it can be seen based on the literature reviewed, that the application of plasma in food processing or in the modification of biopolymers is a novel technique that still requires approval. The Food and Drug Administration (FDA) has not approved any guidelines on the application of plasma in food or on food contact surfaces [120,127]. Research on cold plasma optimization has to be performed specifically for each product. A safety and risk assessment should be comprehensively carried out in both an in vitro and in vivo environment to address the potential toxicity of food or food contact surfaces such as biopolymer films treated with cold plasma.

### 5. Conclusions

The commercial application of biopolymer-based films is limited due to their poor physical, structural, mechanical, thermal, and barrier properties, as well as poor ink printability and adhesion features. The main objective of this review article was to investigate the impact of cold plasma used for the modification of the critical properties of packaging films prepared from protein, polysaccharides, or their combinations. Different factors that influence the effect of plasma treatment include the internal structure of biopolymers, types of plasma gas generating reactive species, and processing conditions (voltage, and treatment time). The application of cold plasma efficiently improved the physical, structural, and thermomechanical properties of the packaging films in most cases. The application of cold plasma modified surface properties enhancing the diffusion rate of functional components absorbed on the surface of the biopolymer. It also enabled adhesion between polymers facilitating the development of multilayer films and increased ink printability. The cold

plasma treatment also enhanced the antimicrobial efficacy by increasing the diffusion rate, and retention of the volatile functional components. In addition, the plasma treatment augmented the biodegradability of the biopolymer-based films. Overall, the application of cold plasma treatment is a cost-effective approach to modifying the packaging properties of biopolymer-based films, as it is a simple inline process with an easy instrumental setup and no waste generation. However, comprehensive research is needed to understand the complex interaction between the plasma reactive species and components of the polymer, as well their migration into food concerning the safety of human and animal health, before cold plasma can be applied commercially.

**Author Contributions:** Conceptualization: S.P., B.K.T. and J.P.K.; writing—original draft preparation, M.H.; writing—review and editing, M.H., C.M., S.P., B.K.T. and J.P.K.; supervision, S.P., B.K.T. All authors have read and agreed to the published version of the manuscript.

**Funding:** This work was funded by Teagasc Walsh Fellowship program, Department of Agriculture, Food, and Marine (DAFM) under the Food Institutional Research Measure (FIRM) program (Pectipack 2019R428) and Science Foundation Ireland (SFI) under grant number 17/CDA/4653.

**Institutional Review Board Statement:** Not applicable.

**Informed Consent Statement:** Not applicable.

**Data Availability Statement:** Data sharing not applicable.

**Conflicts of Interest:** The authors declare no conflict of interest.

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
