# Peer review of "Effect of Cold Plasma Treatment on the Packaging Properties of Biopolymer-Based Films: A Review"

_applsci, doi:10.3390/app12031346_

Round 1

Reviewer 1 Report

Dear authors,

Congratulations for the interesting review.

Food packing is very important more then ever these days. With increasing demand and stricter conditions imposed by current conditions, the food packaging industry is under pressure. The materials used for packaging the food must ensure their preservation for a long time, ensure freshness, do not interact with food, and can be easily recycled / reused. The properties of plastics have made them the main material used in the food packaging industry. Unfortunately, over the years the recycling of this packaging has been out of control for various reasons. Also, some of these plastics have proven to be unsuitable for such uses.

Measures need to be taken to reduce pollution caused using plastic as packaging, and to use more environmentally friendly materials that are easier to recycle.

This review highlights some aspects of using biopolymer films and some treatment methods to improve their use.

A few remarks a I must do:

  1. Most of pictures are taken from literature; you have indicated the source, but did you check if it is not necessary approval from the sources?
  2. Text in some pictures is barely visible – like the last one of figure 2, figure 3b
  3. Why did you used figure 3a and 3b and not figure 3, and figure 4
  4. You have cited some papers using the first and the second author, why not: “name of first author et al.”, or “name of first author and his colleagues”, “name of the first author and his team” ?
  5. Source 100 (line 302) is used before sources 92, 93 …
  6. Lines 762 and 763 – please clarify “…that requires still requires approval”
  7. Table 2 - for Zein, polylactic acid - What is (Td) - why it is specified in that row, and specify after the table what it means, and for the same raw deltaH is blank. Explain after the table the signs: what means "=", "--", arrows

Reviewer 2 Report

The authors well described about cold plasma treatment of the biobased-films. The contents are well orgnized and summarized throughout the manuscript and I am sure this paper will be of interest to a broad reader. However, Table 1 and Table 2 are too difficult to read. I think if the authors can make the Tables portrait with concise contents, that would be easier to read.

I strongly recommend publication in Applied Sciences, as is.

Reviewer 3 Report

The work:  Effect of Cold Plasma Treatment on the Packaging Properties of 1 Biopolymer-Based Films: a Review, presented by Monjurul Hoque, Ciara McDonagh, Brijesh K. Tiwari, Joseph P. Kerry and Shivani Pathania, it is a good review about the subject in nowadays. The schematic figures are useful in comprehension of the work, the references are also extensive on the subject.

As known, the specific physical and chemical properties of the plasma treatment are complex but, in this review, there are some examples for different polymers, which may be helpful.

We have only three remarks:

1) The standard deviation (SD) of the RRMS obtained by AFM are missing in the texts, although it is present in the bar chart Figure3A. In my opinion, May the authors report these SD values in the text?

2) Authors talk about diffusion in L254-255 and it is associated to film roughness. However, diffusion is a bulk property. I believe that the adsorption can also play an important role in this explanation.

3) Finally, the axes and labels in the graphics are so small, see specially Figure 3A y 3B.
